# Investigating the Effects of Seizures on Procedural Memory Performance in Patients with Epilepsy

**DOI:** 10.3390/brainsci11020261

**Published:** 2021-02-19

**Authors:** Frank J. van Schalkwijk, Walter R. Gruber, Laurie A. Miller, Eugen Trinka, Yvonne Höller

**Affiliations:** 1Christian Doppler Medical Centre and Centre for Cognitive Neuroscience, Department of Neurology, Paracelsus Medical University, 5020 Salzburg, Austria; frankjasper.vanschalkwijk@sbg.ac.at (F.J.v.S.); e.trinka@salk.at (E.T.); 2Centre for Cognitive Neuroscience (CCNS), Department of Psychology, University of Salzburg, 5020 Salzburg, Austria; walter.r.gruber@sbg.ac.at; 3Institute of Clinical Neurosciences, Royal Prince Alfred Hospital and Central Medical School, University of Sydney, Camperdown, Sydney, NSW 2050, Australia; laurie.miller@sydney.edu.au; 4ARC Centre of Excellence in Cognition and Its Disorders, University of Sydney, Camperdown, Sydney, NSW 2006, Australia; 5Faculty of Psychology, University of Akureyri, 600 Akureyri, Iceland

**Keywords:** epilepsy, motor learning, offline consolidation

## Abstract

Memory complaints are frequently reported by patients with epilepsy and are associated with seizure occurrence. Yet, the direct effects of seizures on memory retention are difficult to assess given their unpredictability. Furthermore, previous investigations have predominantly assessed declarative memory. This study evaluated within-subject effects of seizure occurrence on retention and consolidation of a procedural motor sequence learning task in patients with epilepsy undergoing continuous monitoring for five consecutive days. Of the total sample of patients considered for analyses (*N* = 53, *M_age_* = 32.92 ± 13.80 y, range = 18–66 y; 43% male), 15 patients experienced seizures and were used for within-patient analyses. Within-patient contrasts showed general improvements over seizure-free (day + night) and seizure-affected retention periods. Yet, exploratory within-subject contrasts for patients diagnosed with temporal lobe epilepsy (*n* = 10) showed that only seizure-free retention periods resulted in significant improvements, as no performance changes were observed following seizure-affected retention. These results indicate general performance improvements and offline consolidation of procedural memory during the day and night. Furthermore, these results suggest the relevance of healthy temporal lobe functioning for successful consolidation of procedural information, as well as the importance of seizure control for effective retention and consolidation of procedural memory.

## 1. Introduction

Problems with memory retention are frequently observed in and reported by patients with epilepsy [1,2], and can considerably affect their overall quality of life [3,4]. Despite the observation that initial assessment of memory performance following learning sometimes shows no differences between patients with epilepsy and healthy participants, long-term memory retention (that is, over hours or days) is often much worse for patients for both semantic and visual material [5,6,7,8]. This long-term forgetting is frequently observed in patients with temporal lobe epilepsy [9,10] and considered more likely to affect the retention of declarative, hippocampus-dependent memory modalities (that is, semantic and episodic memory modalities) compared to procedural memory, which can be considered to be more hippocampus independent [11,12,13]. Yet, recent studies have shown that the consolidation of procedural motor sequence tasks also involves the hippocampus as well as the striatum [14,15] (for a review on procedural memory consolidation, see [16,17]). Furthermore, Schapiro and colleagues have shown that the hippocampus is crucial for the consolidation of a newly learned task without it being involved in the initial acquisition [18]. They evaluated memory performance for a motor sequence task in both hippocampal amnesic patients and matched controls. While performance during the learning session was similar between patients and controls, they found that offline consolidation after a period of sleep resulted in improved performance during memory recall for the control participants only, as no significant performance changes were observed for the patient group. The authors argue that the sequence aspect of the procedural task can be consolidated by the hippocampus, thus resulting in performance improvements in healthy controls [16,19]. Crucially, such hippocampal functioning is limited or lacking in amnesic patients diagnosed with hippocampal damage, which would explain the lack of a performance gain as observed in healthy controls.

Considering the importance of hippocampal functioning for memory consolidation [20,21], any electrophysiological interference can be detrimental for memory retention and performance [22,23]. With the medial temporal lobe being a predilection site for epilepsy, the potential for such interference with hippocampal functioning is high. Consequently, it is believed that electrophysiological interference of hippocampal functioning in the form of epileptiform activity can be detrimental for memory consolidation processes; therefore leading to worsened performance in this patient population [24,25]. In support of this theory, memory retention has been negatively associated with the occurrence of epileptiform activity (i.e., seizures and epileptic discharges), as their onset following learning resulted in poorer retention and recollection of declarative information [5,6,26,27,28,29]. The question is whether seizure occurrence has a similar detrimental effect on the retention and consolidation of procedural memory. 

Although the effects of epileptiform activity on declarative memory performance are frequently studied in patients with epilepsy (for example, [5,6,7,8,26,30,31]), the effects on procedural memory performance are relatively understudied in this patient population. One of the few studies that investigated procedural memory performance in this patient population was conducted by Long and colleagues, who contrasted performance on a motor sequence learning task between patients with epilepsy and healthy adults [32]. Results showed that, although baseline performance was similar between patients and controls, performance improvements from learning to recall were stronger for the healthy controls as compared to the patients. This contrast in performance improvements between patients and healthy controls is similar to results observed in hippocampal amnesic patients [18]. Unfortunately, this study did not directly look at the relation between seizure occurrence and memory retention. Importantly, these results indicate that it is not the initial acquisition of motor sequence tasks that is affected in these patient populations, but rather the subsequent consolidation processes that involve the medial temporal lobe structures and specifically the hippocampus. 

Taken together, these findings illustrate the detrimental effects of epileptiform activity on memory consolidation as well as the relevance of healthy hippocampal functioning for both declarative and procedural memory consolidation. Furthermore, these results suggest that epileptiform activity may negatively affect hippocampal functioning related to memory consolidation of procedural motor sequence learning; potentially preventing any offline performance gains as seen in healthy participants. Yet, a direct link between procedural memory performance changes and epileptiform activity is lacking from the current literature. 

The ability to directly study the effects of a seizure on memory retention in an experimental setting is challenging considering the unpredictability of a seizure occurrence. Consequently, many studies evaluated memory retention following learning after a short (30 min) and long-term (12 h, 24 h, 72 h, or 7 days) retention period, and conducted between-patient contrasts based on the presence or absence of seizures during the retention period. Yet, the within-patient contrasts are of highest interest to study the effects of seizures on memory retention and performance, as the high between-subject variability can be problematic confounds for between-subject analyses (for example, age, epilepsy type, hippocampal atrophy, age of epilepsy onset, as well as the type, number, and dosage of antiepileptic medication). 

The current study aimed to make within-subject evaluations of the effects of seizure occurrence on memory retention and performance changes. Specifically, within-subject contrasts were conducted to determine the effect of seizure occurrence during the retention period separating learning and recall on procedural memory performance. We hypothesized that memory retention and performance changes for procedural memory would be negatively affected by seizure-affected retention periods, whereas offline gains in performance from learning to recall were expected in seizure-free retention periods as observed in healthy participants. 

## 2. Materials and Methods

### 2.1. Patients

A total of 106 consecutive patients were recruited for this study; all of whom underwent continuous monitoring for five consecutive days (Monday-Friday) for diagnostic purposes in the Epilepsy Monitoring Unit (EMU) of the Department of Neurology, Christian Doppler Medical Center, Salzburg, Austria. Patients were excluded from further analyses based on pilot participation (*n* = 21), lack of a clear epilepsy diagnosis (*n* = 17), left-hand preference (*n* = 7), or incomplete behavioral data (*n* = 1). To ensure that our analyses were not influenced by either incomprehension or an inability to perform our procedural memory task, sessions were excluded when no clear learning performance was observed at the end of each learning session (accuracy ≤ 50% or speed ≤ 5), which led to the exclusion of 7 patients. The final sample consisted of 53 right-handed patients who were diagnosed with epilepsy by specialized neurologists. 

### 2.2. Study Protocol

Patients arrived at the EMU on Monday morning, after which they were fully briefed on the study, provided written informed consent, and were screened for depression and chronotype (see Appendix A for additional screening details). The study protocol started on Monday evening and lasted until Thursday evening, with sessions taking place during the morning (07:00–09:00) and evening (18:00–20:00); therefore there were seven sessions in total (Figure 1). The current study was part of a larger investigation that aimed to evaluate the effect of seizures on the consolidation of procedural, semantic, and episodic memory. Therefore patients were trained and evaluated on (1) procedural memory, (2) verbal memory, and (3) visuospatial memory for each session (for details on visuospatial memory, see [33,34]). Learning performance for the procedural memory task was determined at the end of the learning session (average of last three trials), whereas learning performance on verbal and visuospatial memory was evaluated through immediate recall. Delayed recall performance was evaluated for all tasks at the start of the subsequent session, after which patients were once again trained and evaluated on a new version for each task (for details see Appendix A). This paradigm was repeated until the last session, during which patients only had to recall the information acquired during the previous session. The study protocol was approved by the Ethics Commission Salzburg (E/1755). 

### 2.3. Instruments

#### 2.3.1. Epilepsy Monitoring 

The EMU consists of four beds that are situated in one room for simultaneous observation of four patients through video and electroencephalography recordings (EEG; Micromed Brain-Quick System). Continuous 24 h monitoring was conducted by medical staff situated in an adjacent room. Daily rhythm was roughly standardized for all patients, with specific time ranges for lights on (06:30–07:00), breakfast (07:00–07:30), lunch (11:30), dinner (16:30), and lights off (22:00–00:00).

#### 2.3.2. Procedural Memory-Motor Sequence Learning Task 

Patients were trained and evaluated on procedural memory performance using a fingertapping task [35] (Figure 2). Patients were instructed to continuously type a specific five-element sequence made of four numbers (sequences used: 1–4–2–3–1, 4–1–3–2–4, 3–2–1–4–3, 2–1–3–4–2, 1–2–4–3–1, and 4–3–1–2–4) as quickly and accurately as possible with their non-dominant left hand. Four keyboard keys were assigned to the four numbers that had to be typed using index-to-little fingers, respectively. Both learning and recall sessions always displayed the desired sequence in the center of the screen and did not provide feedback regarding performance. Each trial lasted 30 sec during which the patient was encouraged to repeatedly type the sequence as quickly and accurately as possible. Trials were separated by an inter-trial interval (ITI) of 30 sec. The learning sessions consisted of 12 trials whereas recall sessions consisted of 4 trials. Performance was evaluated based on speed (number of completed sequences per trial), triplets (number of correct 3-element inputs belonging to the desired sequence (i.e., for sequence 1–4–2–3–1, triplets are 1–4–2, 4–2–3, 2–3–1, 3–1–1, and 1–1–4), and accuracy (percentage of elements belonging to correct sequences relative to total keystrokes per trial). For analyses, learning and recall performances were determined by averaging the last three trials of their respective session. Performance changes (recall-learning) were calculated for each performance measure.

### 2.4. Analyses

Seizures were identified, marked, and classified as “tonic-clonic” seizures or “other” by trained medical staff. Behavioral data were processed using Matlab v9.5 (R2018b; Natick, MA, USA) and R v3.5.1. Statistical comparisons were conducted using IBM SPSS Statistics v22 (Armonk, NY, USA) and R v3.5.1. Potential confounding factors such as age, time of day, task difficulty, and day within the paradigm were initially investigated. Behavioral performance was contrasted based on the type of retention interval (daytime vs. nighttime) and the occurrence of seizures during retention. Note that nights during which sleep deprivation was applied were excluded from analyses. Within-patient contrasts were conducted for subsamples extracted from the final sample of patients (*N* = 53) depending on the contrasts in question (that is, seizure-free versus seizure-affected retention periods; seizure-free daytime versus nighttime retention periods). Behavioral performance was averaged per patient for learning and recall sessions respective to their retention period. Main and interaction effects were investigated using 2 × 2 (TIME × CONDITION) repeated-measures analyses of variance (ANOVAs) using the “RM” function from the “MANOVA.RM” package [36] using 10,000 iterations by the parametric bootstrap approach. This approach is recommended considering we conducted both within- and between-subjects design characterized by low as well as unequal sample sizes, respectively. Within-patient contrasts were conducted using Wilcoxon signed rank tests that were corrected for multiple comparisons using the Bonferroni-Holm procedure for the number of tests conducted. Effect size estimates and 95% confidence intervals (CI) were calculated in R using the “effsize” package [37]. Results reported mean ± standard deviation unless otherwise specified.

## 3. Results

### 3.1. Patient Demographics and Baseline Descriptive

The final sample consisted of 53 right-handed patients (M_age_ = 32.92 ± 13.80 y, range = 18–66 y; 43% male). Of these 53 patients, 17 patients (32%) experienced one or more seizures during monitoring (daytime seizures only, *n* = 6; nighttime seizures only, *n* = 7; daytime and nighttime seizures, *n* = 4). Patients’ descriptive information is displayed in Table 1. Further patient details regarding epilepsy diagnosis, localization, lateralization, and medication are reported in Appendix A. 

### 3.2. Confounding Factors

Potential confounding factors such as age, task difficulty, learning effect, and circadian effects were evaluated. As the main focus of this study lies on the relative change in performance from learning to delayed recall conditions within patients, and therefore does not specifically account for these factors (for a more detailed discussion, see Appendix A). Patients’ medication usage was tailored to individual requirements and therefore varied strongly (see Appendix A). To increase the occurrence of epileptiform activity during the monitoring, medication was tapered in 60% of patients. Tapering was implemented based on the individual patient history and consequently varied between patients. Analyses therefore did not take medication or its dosage into account.

### 3.3. Seizure Occurrence

A subset of patients (*n* = 17) experienced one or more seizures during their stay in the EMU. Of these, behavioral data for both seizure-free and seizure-affected retention periods were available from 15 patients for within-subject analyses. Retention periods were classified based on the presence or absence of seizures (seizure-free daytime retention (*n_sessions_* = 219, *n* = 52); seizure-free nighttime retention (*n_sessions_*=215, *n*=53); seizure-affected daytime retention (*n_sessions_* = 12, *n* = 10); and seizure-affected nighttime retention (*n_sessions_* = 16, *n* = 11)). 

### 3.4. Behavioral Within-Subject Contrasts 

#### 3.4.1. Seizure-Free vs. Seizure-Affected Retention Periods

Within-patient contrasts were conducted to compare retention periods with and without seizures (*n* = 15). Performance for each category of retention period (that is, with or without seizures) was averaged per patient for learning and recall sessions. We conducted separate 2 × 2 repeated-measures ANOVAs (TIME × CONDITION) to evaluate performance changes for speed, triplets, and accuracy. Results showed a main effect of time for speed (*F*_1,14_ = 4.844, *p* = 0.042, $\eta_{G}^{2}$ = 0.017), whereas no main effect of condition *(F*_1,14_ = 2.493, *p* = 0.134), nor an interaction effect was observed (*F*_1,14_ = 0.302, *p* = 0.593). Similarly, triplets showed a main effect of time (*F*_1,14_ = 5.743, *p* = 0.029, $\eta_{G}^{2}$ = 0.017) and a nonsignificant trending main effect of condition (*F*_1,14_ = 3.187, *p* = 0.092), whereas no interaction effect was observed (*F*_1,14_ = 0.613, *p* = 0.441). For accuracy, no main effects nor an interaction effect were observed (all *p* ≥ 0.646). Additional direct contrasts are reported (Table 2). Both learning and recall performance scores were similar when retention periods with and without seizures were compared. When investigating performance changes from learning to recall, retention periods without seizures resulted in a significant performance improvement for speed (*d* = −0.951, 95% CI (−1.740, −0.162]) and a trending improvement for triplets (*d* = −0.954, 95% CI (−1.743, −0.165)). In contrast, no changes were observed for any performance measure following retention periods with seizures. Crucially, fatigue did not differ within the seizure-free condition between learning (3.61 ± 1.77) and recall sessions (3.68 ± 1.53; *Z* = −0.746, *p* = 0.46), nor within the seizure-affected condition between learning (4.45 ± 2.27) and recall sessions (4.33 ± 2.16; Z = −0.140, *p* = 0.89), nor were there any differences in fatigue between the seizure-free and seizure-affected conditions during learning (*Z* = −1.399, *p* = 0.16) and recall (*Z* = −1.190, *p* = 0.23). 

#### 3.4.2. Seizure-Free Day vs. Seizure-Free Night 

Retention periods of days and nights without seizures were contrasted within patients (*n* = 42). Performance for each category of retention period (that is, daytime or nighttime) was averaged per patient for learning and recall sessions. We again conducted separate 2 × 2 repeated-measures ANOVAs (TIME × CONDITION) for speed, triplets, and accuracy. For speed, we found a main effect of time (*F*_1,41_ = 9.677, *p* = 0.003, $\eta_{G}^{2}$ = 0.007) and a nonsignificant trend of condition (*F*_1,41_ = 3.490, *p* = 0.066, $\eta_{G}^{2}$ = 0.003), whereas no significant interaction was observed (*F*_1,41_ = 0.009, *p* = 0.928). Similarly, triplets showed a main effect of time (*F*_1,41_ = 14.520, *p* < 0.001, $\eta_{G}^{2}$ = 0.01), whereas no effect of condition (*F*_1,41_ = 2.587, *p* = 0.118) nor an interaction effect was observed (*F*_1,41_ = 0.007, *p* = 0.931). Finally, no main nor interaction effects were observed for accuracy (all *p* ≥ 0.280). Additional direct contrasts are reported (Table 3; Figure 3). Contrasting performance for each condition showed no difference between seizure-free daytime and nighttime retention periods for any performance measure. When investigating specific performance changes from learning to recall, seizure-free days showed a trend towards improved performance for triplets (*d* = −0.469, 95% CI (−0.910, -0.029)), whereas seizure-free nights showed improvements in performance that were significant for triplets (*d* = −0.453, 95% CI (−0.892, −0.013)) and trending for speed (*d* = −0.344, 95% CI (−0.781, 0.094)). Importantly, contrasting performance changes from learning to recall between seizure-free days versus seizure-free nights showed no significant differences. Also, self-reported indications of fatigue did not differ within the seizure-free day condition between learning (3.52 ± 2.28) and recall sessions (3.40 ± 1.80; *Z* = −0.083, *p* = 0.93), while the seizure-free night condition showed a trend towards slightly lower fatigue for learning in the evening (3.17 ± 1.67) compared to recall the subsequent morning (3.74 ± 2.14; *Z* = −1.919, *p* = 0.06). Crucially, no differences in fatigue was observed between the seizure-free day and night conditions during learning (*Z* = −1.057, *p* = 0.29) and recall (*Z* = −0.747, *p* = 0.46).

### 3.5. Exploratory Within-Subject Contrasts in Patients with Temporal Lobe Epilepsy

Considering that patients with temporal lobe epilepsy showed significantly worse long-term memory retention [9,10] as well as lower performance changes on procedural memory as compared to healthy controls [32], we conducted an exploratory within-subjects contrast specifically for those patients who experienced seizures during the paradigm and were diagnosed with epilepsy originating from the temporal lobe (*n* = 10; see Appendix A). As with the previous contrasts, we found main effects of time for the speed (*F*_1,9_ = 5.448, *p* = 0.020, $\eta_{G}^{2}$ = 0.010) and triplets (*F*_1,9_ = 6.156, *p* = 0.013, $\eta_{G}^{2}$ = 0.010) variables, whereas no main effects were observed of condition (all *p* ≥ 0.368). Yet, we observed significant interaction effects (TIME × CONDITION) for speed (*F*_1,9_ = 8.017, *p* = 0.021, $\eta_{G}^{2}$ = 0.007) as well as triplets (*F*_1,9_ = 9.772, *p* = 0.015, $\eta_{G}^{2}$ = 0.006) (Figure 4). Post hoc tests revealed that the seizure-free retention periods resulted in higher improvements of performance for speed (1.937 ± 1.238; *p* = 0.020) and triplets (9.855 ± 7.653; *p* = 0.012) as compared to the seizure-affected retention periods (0.178 ± 2.122 and 1.141 ± 8.859, respectively). No significant main or interaction effects were observed for accuracy (all *p* ≥ 0.495).

### 3.6. Between-Subjects Contrasts

We conducted additional 2 × 2 repeated-measures ANOVAs (TIME × GROUP), contrasting patients who experienced seizures (*n* = 15) with those who did not (*n*=36). Note that we again contrasted average patient performance between groups on sessions that did not involve the occurrence of a seizure. As with the within-subjects contrasts, we found main effects of time for the variables speed (*F*_1,49_ = 15.456, *p* < 0.001, $\eta_{G}^{2}$ = 0.006) and triplets (*F*_1,49_ = 17.984, *p* < 0.001, $\eta_{G}^{2}$ = 0.008), whereas no main effect of group nor any interaction effects were observed (all *p* ≥ 0.091). 

## 4. Discussion

### 4.1. General Discussion

This study investigated the direct effects of seizures on procedural memory performance in patients with epilepsy. Our results showed that patients were able to significantly improve their performance on a procedural motor sequence learning task despite the lack of additional training. Importantly, no within-subject differences were observed in performance improvements depending on whether patients experienced a seizure or not during the retention period separating learning and recall (*n* = 15). However, an exploratory within-subject contrast specifically for patients diagnosed with temporal lobe epilepsy (*n* = 10) showed that performance improvements were significantly better after a seizure-free retention period as compared to a seizure-affected retention period. Furthermore, we observed no differences between performance changes following a period of daytime wakefulness or nighttime sleep; suggesting that the acquired information can be consolidated during both retention periods, thus resulting in performance improvements during recall.

The lack of a general effect of seizure occurrence on procedural memory performance is striking given the results of prior studies on the negative effects of epileptiform activity on declarative memory consolidation [5,29,38]. Yet, the exploratory within-subject contrast, which was specific for patients diagnosed with temporal lobe epilepsy, did illustrate the negative effects of seizure occurrence on procedural memory performance by preventing offline performance improvements. Specifically, the lack of procedural performance changes following a seizure is a result directly in line with previous findings [18,32] and suggests the importance of healthy temporal lobe functioning for procedural memory consolidation. Importantly, it should be noted that none of the within-patient contrasts showed a main effect of condition. In other words, these results do not support the statement that seizure occurrence is detrimental for procedural memory performance, but rather illustrate how seizure occurrence following learning can be detrimental for potential performance gains through offline consolidation during the subsequent retention period. Yet, we must stress the exploratory nature of this within-subject contrast as well as the low sample size and associated effect sizes. The within-patient contrast on seizure-free retention periods during the day and night suggests that offline improvements in procedural memory consolidation can occur during both time points, whereas previous studies specifically emphasize the importance of nighttime sleep for procedural memory consolidation, as demonstrated in healthy participants [16] and patients with epilepsy [39,40]. Importantly, some patients reported taking daytime naps in the afternoon as a result of their restricted mobility within the EMU (that is, bed restricted and limited in their actions and movements). Thus, it is possible that this daytime improvement is a result of sleep during the daytime retention period, but could also be a simple lack of daytime distractions inherent to the restricted mobility within the EMU. Although a direct within-patient contrast between daytime wake and daytime nap retention periods had insufficient power to draw statistical conclusions, the benefit of a daytime nap for the consolidation of procedural motor sequence learning has been established for healthy participants [16,41,42], and may be a way to foster memory consolidation in patients with epilepsy to ameliorate memory problems [5], thus aiming to improve their overall quality of life [3,4]. 

### 4.2. Limitations

Despite the use of an elaborate paradigm to evaluate the effects of seizures on procedural memory consolidation, the current study must acknowledge several limitations. First and foremost, we acknowledge the limits of the low sample size, which is a direct result of the unpredictability of seizure occurrence. Second, although the EMU offers the opportunity to conduct a long-term study in patients with epilepsy, it is not the ideal experimental environment (for example, noise and interference from other patients, medical staff, and hospital activities). Third, while the occurrence of seizures was promoted through tapering of medication, this was not implemented for all patients. Furthermore, tapering of medication could potentially have affected cognitive performance [43]. In addition, the specific (combination of) medication and its dosage varied between patients and could not be controlled or standardized. Fourth, the negative effect of interictal epileptiform discharges (IEDs) was not investigated in the current study, but is a topic of high relevance in the context of memory retention and cognitive performance [44,45]. Fifth, post-ictal fatigue frequently resulted in the (temporary) withdrawal from the study, leading to a loss of data and a relatively low number of samples that could be used for within-subject analyses. This also forced us to omit any further investigations that differentiated between patients based on their seizure type. Sixth, the three memory tasks might potentially have interfered with each other [46]. For this reason the tasks were kept in a consistent non-randomized order for all patients. Seventh and last, a detailed study on the interplay between epilepsy, memory consolidation, and sleep is of high relevance [24,25]. Specifically, previous research has indicated that epileptiform discharges may result in an alteration of both sleep architecture [47,48,49] and sleep microstructures [22,24,50], which is likely to underlie some of the memory problems in patients with epilepsy [24,25,51,52]. Unfortunately, such a direct within-patient contrast was not possible given the low number of patients who experienced seizures during the night (*n* = 11). Despite these limitations, we would like to stress that this study, through its continuous 5-day recording time, had the unique opportunity to evaluate performance changes on procedural memory following a seizure.

## 5. Conclusions

Our study shows that procedural memory performance generally improved following daytime and nighttime retention periods, as well as following retention periods that were seizure-free or seizure-affected. Yet, specific contrasts for patients diagnosed with temporal lobe epilepsy showed that improvements for procedural memory performance only occurred during seizure-free retention periods, whereas no performance changes were observed following seizure-affected retention periods. These findings suggest the relevance of healthy temporal lobe functioning for the consolidation of procedural information. These results contribute to previous observations on the negative effect of seizures on memory retention and stress the relevance of seizure control for the benefit of memory retention in patients with epilepsy.

## Figures and Tables

**Figure 1 brainsci-11-00261-f001:**
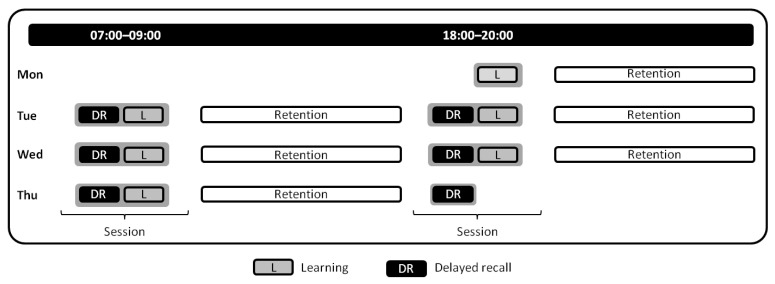
Visual representation of the study protocol. The figure illustrates the study protocol specifically for the procedural memory task (for an overview of the full study protocol including verbal and visuospatial memory tasks, see Appendix A). The first session took place on Monday evening and included a learning session of the procedural memory task. All subsequent sessions except for Thursday evening included delayed recall of the previously learned task version and the learning of a new task version.

**Figure 2 brainsci-11-00261-f002:**
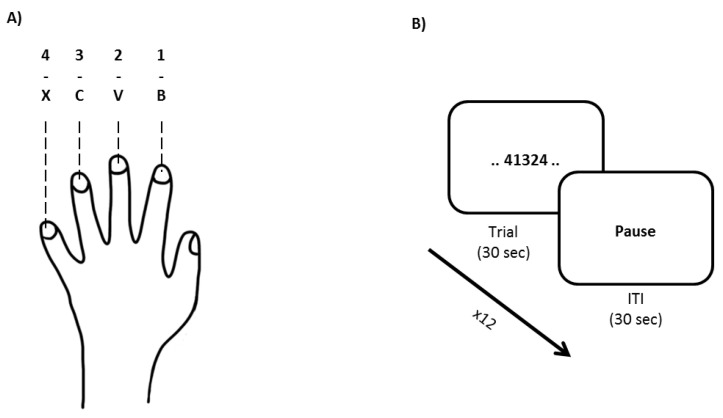
Visualization of the procedural memory task. (**A**) Participants used their non-dominant left hand to type the desired sequence. Each finger was assigned to a specific keyboard key that represented a number of the desired sequence. (**B**) Participants were shown a specific 5-element sequence on a screen and were instructed to type the desired sequence as frequently and accurately as possible for each trial (30 s) over the course of 12 trials; each trial was separated by an inter-trial interval (ITI) of 30 s. Subsequent delayed recall consisted of only four trials.

**Figure 3 brainsci-11-00261-f003:**
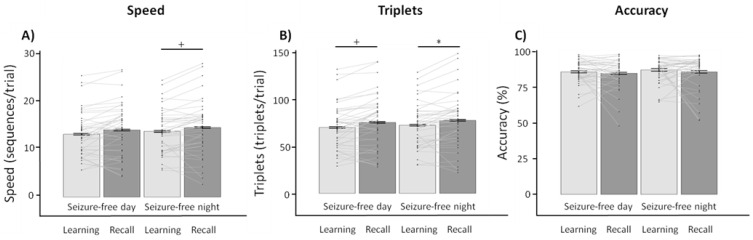
Evaluating performance (changes) following a seizure-free day and seizure-free night (*M* ± *SE*). Note that error bars have been corrected to represent within-subject variability. Contrasting performance between learning and recall sessions following a seizure-free day or night showed significant improvements for (**A**) speed and (**B**) triplets, whereas no changes were observed for (**C**) accuracy. Importantly, the performance changes from learning to recall were similar for seizure-free days and nights for speed, triplets, and accuracy. ^+^
*p* < 0.100; * *p* < 0.050.

**Figure 4 brainsci-11-00261-f004:**
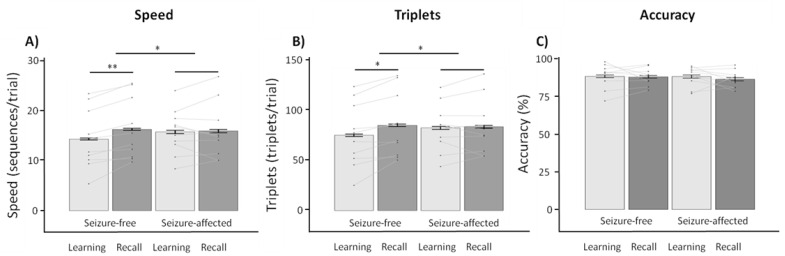
Exploratory within-subject contrasts (*n* = 10) following a seizure-free and seizure-affected retention period for patients with temporal lobe epilepsy (*M* ± *SE*). Note that error bars have been corrected to represent within-subject variability. Contrasting performance between learning and recall sessions following a seizure-free or seizure-affected retention period (**A**–**C**) showed performance improvements from learning to recall for speed and triplets. Furthermore, performance improvements were higher for a seizure-free as compared to a seizure-affected retention period. * *p* < 0.050; ** *p* < 0.001.

**Table 1 brainsci-11-00261-t001:** Descriptive information on demographic and epilepsy.

Demographics	
Age (*M* ± *SD*)	32.92 ± 13.80
Gender (M/F)	23/30
**Epilepsy characteristics**	
**Epilepsy Type**	
Focal	43
Generalized	10
**Location**	
Frontal	8
Temporal	14
Occipital	2
Bifrontal	2
Fronto-central	1
Fronto-temporal	5
Fronto-centro-parietal	1
Temporo-mesial	3
Parieto-occipital	1
Unclear	16
**Lateralization**	
Left	20
Right	12
Bilateral	8
Unclear	13

**Table 2 brainsci-11-00261-t002:** Within-patient performance changes evaluated for retention periods based on the absence or presence of seizures (*n* = 15).

		Learning	Recall		
Speed		*M ± SD*	*M ± SD*	*Z*	*p*
Seizure-free retention		14.61 ± 5.14	16.10 ± 5.43	−2.897	0.048 *
Seizure-affected retention		15.65 ± 3.84	16.66 ± 5.29	−0.816	1.000
	*Z*	−1.619	−1.108		
	*p*	0.954	1.000		
**Triplets**					
Seizure-free retention		76.66 ± 27.38	84.78 ± 28.17	−2.811	0.055 ^+^
Seizure-affected retention		82.42 ± 20.36	87.48 ± 26.36	−0.909	1.000
	*Z*	−1.675	−1.221		
	*p*	0.940	1.000		
**Accuracy**					
Seizure-free retention		88.21 ± 7.14	87.16 ± 6.56	−0.426	1.000
Seizure-affected retention		87.35 ± 6.28	87.44 ± 5.22	−0.085	1.000
	*Z*	−0.628	−0.199		
	*p*	1.000	1.000		

Note: *p*-values were corrected for multiple comparisons using the Bonferroni–Holm procedure (for all comparisons (*n* = 12)). ^+^
*p* < 0.100; * *p* < 0.050.

**Table 3 brainsci-11-00261-t003:** Within-patient performance changes evaluated for retention periods that included Scheme 42.

		Learning	Recall		
Speed		M ± SD	M ± SD	*Z*	*p*
Seizure-free retention		13.18 ± 4.77	14.03 ± 5.50	−2.093	0.324
Seizure-affected retention		13.77 ± 4.55	14.58 ± 5.67	−2.732	0.066 ^+^
	*Z*	−1.663	−1.309		
	*p*	0.672	0.905		
**Triplets**					
Seizure-free retention		69.35 ± 24.10	74.53 ± 28.20	−2.651	0.080 ^+^
Seizure-affected retention		71.75 ± 23.23	76.75 ± 28.03	−3.282	0.012 *
	*Z*	−1.200	−1.575		
	*p*	0.905	0.690		
**Accuracy**					
Seizure-free retention		85.47 ± 7.96	84.42 ± 9.95	−0.313	1.000
Seizure-affected retention		86.86 ± 8.12	85.50 ± 11.77	−0.025	1.000
	*Z*	−1.901	−1.338		
	*p*	0.456	0.905		

Note: *p*-values were corrected for multiple comparisons using the Bonferroni–Holm procedure (for all comparisons (*n* = 12)). For visualization of the data, see Figure 4. ^+^
*p* < 0.100; * *p* < 0.050.

## Data Availability

Data is available on request due to restrictions (for example, privacy or ethical). The data presented in this study are available on request from the corresponding author.

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
