# Peer review of "Investigating the Effects of Seizures on Procedural Memory Performance in Patients with Epilepsy"

_brainsci, 2021, doi:10.3390/brainsci11020261_

Round 1
Reviewer 1 Report
In this manuscript Schalkwijk and colleagues report on within-subject effects of epileptic seizures on the retention of motor sequence learning in a multi-session design. The main finding of this study is a significant improvement of sequence memory during retention intervals without seizures, whereas there is no such improvement for retention intervals with seizures. While the within-subject analysis can certainly add to the current literature which mostly reports between-subject effects, the current statistical analysis does not allow for any of the conclusions drawn by the authors, neither for the seizure vs. no-seizure nor for day vs. night findings. Also, there are some issues with the implication of certain neural correlates mentioned in the introduction.
- The authors imply a difference between memory retention for intervals with vs. without seizures, however from a significant pre-post comparison within no-seizure sessions and a non-significant comparison within seizure-sessions this cannot be concluded. A statistical test comparing performance in no-seizure vs. seizure sessions needs to be done.
- Along the same lines, for the comparison of performance change over seizure-free retention intervals during the day vs. the night the analysis mentioned above was done, yielding no significant difference. Still, in the discussion it is stated, that performance improvements were mainly achieved during night retention intervals. This needs to be corrected.
- Why do the authors not additionally report on between-subject effects? Considering the large number of patients without seizures it seems like a shame to just exclude these data. Perhaps this would even allow for a matching of important confounds like age in the two groups.
- In the introduction, the authors elaborate on the connection between hippocampal function and procedural memory. While these are interesting findings, and seizures disrupting hippocampal functioning could add to the existing literature, the current dataset is of limited suitability to corroborate this claim, since more than one third of included patients does not have the focus of epileptiform activity in the temporal lobe.
- To actually disentangle the effects of seizure occurrence and time of day of the retention interval the two factors need to be included into one analysis. I realize that the number of data points per factor combination is quite low, however mixed models can cope with the missing values.
Reviewer 2 Report
In this manuscript, van Schalkwijk and colleagues investigated the effect of seizures on procedural memory consolidation in 53 patients with epilepsy during long-term video-EEG recording. They mainly performed pairwise analysis between seizure-free and seizure-affected period for leaning procedure in each patient (N=15 for this analysis). As a result, seizure-free periods resulted in better procedural memory consolidation compared with seizure-affected periods. The authors emphasize the importance of seizure control and seizure-free sleep for consolidation of procedural memory.
Overall, this study is very interesting and high-quality using systematically collected data and careful analysis and methodologies. The manuscript is also well-written. I have relatively minor comments below.
- One potential limitation could be the interval of sessions. Patients underwent the learning and recall sessions twice a day, while PWE sometimes experience memory problems over much longer period, e.g., days, weeks, months. Given the literature on accelerated long-term forgetting and autobiographical memory loss in PWE, further studies investigating long-term memory functions might be desirable in the future.
- According to the Method section, the authors used Bonferroni’s correction for multiple comparisons. Each Table 2 and 3 contains 6 comparisons, and then how many comparisons did the authors correct? 6? Or 12? Please just specify it.
- The burden of memory dysfunction on PWE’s quality of life should be briefly introduced in Introduction (or Discussion).
- Line 63-64, “… lacking in patients amnestic patients diagnosed with …”. Typo?
Round 2
Reviewer 1 Report
The authors have thoroughly tended to all my comments.